# An Atypical Case of Aphasia: Transitory Ischemic Attack in a 13-Year-Old Patient with Asymptomatic SARS-CoV-2 Infection

**DOI:** 10.3390/children9070983

**Published:** 2022-06-30

**Authors:** Marco Scaglione, Flavia Napoli, Giulia Prato, Mariasavina Severino, Marta Bertamino, Sara Signa, Mohamad Maghnie

**Affiliations:** 1Department of Neuroscience, Rehabilitation, Ophthalmology, Genetics and Maternal-Child Sciences (D.I.N.O.G.M.I), University of Genoa, 16132 Genoa, Italy; mohamadmaghnie@gaslini.org; 2Integrated Department of Paediatric and Hemato-Oncological Sciences, IRCCS Istituto Giannina Gaslini, 16147 Genova, Italy; flavianapoli@gaslini.org; 3Integrated Department of Medical and Surgical Neuroscience and Rehabilitation-Continuity of Care, IRCCS Istituto Giannina Gaslini, 16147 Genoa, Italy; giuliaprato@gaslini.org; 4Neuroradiology Unit, IRCCS Istituto Giannina Gaslini, 16147 Genoa, Italy; mariasavinaseverino@gaslini.org; 5Physical Medicine and Rehabilitation Unit, IRCCS Istituto Giannina Gaslini, 16147 Genoa, Italy; martabertamino@gaslini.org; 6COVID Hospital, IRCCS Istituto Giannina Gaslini, 16147 Genoa, Italy; sarasigna@gaslini.org

**Keywords:** pediatric SARS-CoV-2 infection, neurological complications, aphasia, stroke

## Abstract

We report the case of a 13-year-old patient, female, born in Northern Italy, who presented with an acute episode of aphasia, lasting about 15 min, accompanied by left arm dysesthesia. The state of consciousness remained preserved throughout the episode. After a first clinical evaluation at second-level hospital, the patient was sent to our institute for further investigations. Brain MRI performed at admission showed no noteworthy structural alterations. Electroencephalogram was not significant, as was the echocardiographic examination. ECG was normal, except for a corrected-QT at the upper limits of the normal range for age and gender. The neurological examination was substantially normal for the entire duration of the hospital stay. The symptomatology initially described has never reappeared. Blood tests were substantially negative, in particular thrombophilic screening excluded hereditary-familial thrombophilic diseases. Color doppler ultrasound of the supra-aortic trunks, splanchnic vessels and lower limbs were also normal. Only positivity to SARS-CoV-2 serology is reported. In the recent clinical history there were no symptoms attributable to symptomatic coronavirus infection.

## 1. Introduction

It is now known that the complications of SARS-CoV-2 infection do not only affect the pulmonary district. In fact, this is a systemic inflammatory disease that, among the various districts, can also affect the central nervous system. Cases of neurological complications, including cerebrovascular events, are widely described in patients with SARS-CoV-2 infection. Quintanilla-Sanchez et al., in a recent meta-analysis with data on 11,886 patients, reported at least one acute cerebrovascular event in 3.6% of the cases. About the severe form of infection, authors reported an increased generic risk of cerebrovascular events (odds ratio 1.96) but mainly of hemorrhagic stroke (OR 4.12). No statistically significant differences were found regarding the risk of developing ischemic stroke between patients with severe and nonsevere COVID-19 (*p*-value = 0.14) [1]. Nalleballe et al., among 40,469 adult patients, reported a 1% prevalence of stroke and TIA [2]. The main mechanism underlying the neurological complications of the virus is likely attributable to the state of systemic inflammation and hypercoagulability [3,4]. Moreover, according to recently reported cases, in patients with polymorbidity, standard anticoagulation may be insufficient to meet SARS-CoV-2-associated severe hypercoagulability syndrome and prevent the occurrence of cerebrovascular events [5,6]. In the pediatric field this topic is less discussed but published data have been growing recently. In a review published in 2021, Schober et al. reported several cohorts of pediatric patients in which the neurological consequences of SARS-CoV-2 infection were evaluated [7]. Among the various cited, La Rovere et al. reported a neurological manifestation rate of 22% with an ischemic or hemorrhagic cerebrovascular event rate of 0.7% in 1625 SARS-CoV-2 diagnosed patients [8]. Ranabothu et al. found a similar overall frequency of neurological manifestations; in particular headache in 4%, anosmia in 2%, seizures and stroke in 0.7% [9]. In a systematic review published in 2020, Panda et al. reported in pediatric patients a 17% prevalence of non-specific neurologic manifestations (headache, fatigue, and myalgia) and other manifestations such as encephalopathy and meningeal signs in 1% [10]. We should also consider that some neurological manifestations of COVID-19 in adults and children, possibly triggered by autoimmunity, such as Guillain-Barré syndrome, encephalomyelitis, and autoimmune necrotizing myositis, have followed all types of SARS-CoV-2 infection: non-neurological, neurological, and asymptomatic [11]. Furthermore, some authors have found cognitive and emotional (as well as physical) dysfunctions in patients even 6–8 months after SARS-CoV-2 infection [12,13]. In some case reports, authors reported that neuropsychiatric sequelae may also occur in SARS-CoV-2 patients that had not been admitted to the intensive care unit (ICU) [14,15]. Because of the above, it is very important to recognize cases of neurological manifestations of SARS-CoV-2 in order to start a close neurological follow-up in selected cases. In this paper we report the case of a 13-year-old girl, in apparent good health, with neurological symptoms attributable to SARS-CoV-2 infection. 

## 2. Case

M.M., a 13 year-old female, entered our institute on 6 June 2021. Her first evaluation was in Savona Hospital ER about an episode of aphasia and dysesthesia of the left upper limb. The patient’s mother reported the emission of phonemes attributable to semantic paraphasias. These symptoms lasted about 15 min, the state of consciousness was preserved and the patient reported remembering the episode. In the previous days no significant events were reported, except for one episode of right frontal headache about two days before, with spontaneous resolution. The patient had a negative family for headache, neurological, or cardiovascular pathologies. Past medical history did not include significant pathological conditions. The patient was born at term from regular, uneventful pregnancy; delivery was spontaneous. No problem was reported in perinatal period. Recommended vaccinations were regularly performed according to current Italian schedule, and no adverse reaction was reported. Personal history was silent, a part from chickenpox infection at the age of 5 years. Psychomotor milestones including language development were met regularly. No allergies were reported, nor ongoing therapies. She wore corrective lens for myopia. She attended school regularly (final year of middle school) with good grades, and played volleyball at a competitive level. All periodical medical examinations for competitive fitness were found to be normal. Clinical evaluation performed at peripheral hospital failed to detect any significant alteration. The swab for SARS-CoV-2 was weakly positive in the peripheral hospital and it was repeated at our institute, with negative result. In our ER, neuropsychiatric evaluation consult reported normal speech, with no aphasia or dysarthria. Cranial nerve function was apparently preserved, osteotendinous reflexes were normal, Romberg’s and Mingazzini’s tests were negative; sensitivity was preserved in all four limbs, force tests were carried out without alteration. No specific aphasia test was needed due to complete recovery of signs and symptoms. For further investigation, hospitalization was ordered and a brain angio-MRI was required. At the imaging exam, no morpho-structural changes emerged (Figure 1). Urine toxicological screening was performed with negative result. Therefore, electroencephalogram and ophthalmic evaluation (with fundus oculi) were carried out: no pathological alterations were found. EEG showed specific slow waves on right posterior regions. ECG exam was normal, except for heart rate-corrected QT-interval in the upper limit of normal range for age and gender (454 ms). For this reason, 24-h Holter ECG and echocardiogram were required. Echocardiography showed no evidence of alterations in heart contractility. The 24-h ECG-Holter was performed before discharge, with negative results. Routine blood tests (Table 1) were performed, including thrombophilic screening (including genetic analysis for factor V and factor II mutations), anticardiolipin, anti beta2-glycoprotein, and anti-transglutaminase antibodies. Results were essentially normal, except for increase in indirect and total bilirubin (in accordance with the color of the patient’s sclerae, this alteration could be attributed to underlying Gilbert’s syndrome) and SARS-CoV-2 serology which showed a positivity for IgG, indicating a likely previous infection. Moreover, partial IgA deficiency was detected. The study of thyroid function showed no significant alterations. Inflammatory indices were negative. Hereditary thrombophilic diseases were excluded. During hospitalization, the patient’s clinical conditions remained stable. The initial symptoms have not reappeared. For this reason, after re-evaluation of the case, it was not considered necessary to undertake any therapeutic measures, although the use of prophylactical cardio aspirin at a dosage of 3–5 mg per kilo is now well established in pediatric cerebrovascular events. Clinical reassessment was carried out after one and two weeks. In this period of time, no significant symptoms were reported by the patient. Patient gradually resumed her daily activity, including middle school final exams. Neurological objectivity was negative on both occasions. An ECG control was performed with normal results (QTc normalized); also, ecocolordoppler exams of lower limbs, abdomen vessels and supra-aortic trunks were performed, with no evidence of embolic foci.

At clinical follow-up six months after the event, complete clinical well-being was found; no similar critical events or episodes of language disorders were reported by the patient. There were also no reported episodes of visual problems. The patient appeared to be normally oriented in time and space, her speech was fluent—she actually sounded much more brilliant than expected for her age, even though IQ was not measured by standardized tests—and no clinical signs referable to deficits in higher cortical functions were detected. The neurological examination was completely negative; in particular, no motor and sensory deficits were detected on either body side. 

Regarding the neuropsychological aspect, no relics of the event were observed with total maintenance of cognitive functions. The patient regularly started her secondary education.

## 3. Differential Diagnosis and Discussion

Differential diagnosis of this case includes various pathological conditions, including epilepsy. However, in the patient’s clinical history there were no other episodes attributable to seizures, and EEG without specific epileptic abnormalities, together with the long duration of symptoms, do not seem to be indicative for epilepsy. The hypothesis of migraine with aura is unlikely as the patient had never suffered from recurrent headaches, and family history is negative for migraine. A single episode of cephalalgic symptoms, about two days before access to the emergency area, with spontaneous resolution, was reported by the patient. However, this episode did not present the typical characteristics of migraine. The time span between this symptomatology and aphasia episode also makes a correlation unlikely. Conversion disorder is another hypothesis, but the duration of the symptoms does not fit with it. Moreover, the patient did not seem to present the personality structure and psychiatric comorbidities that are normally present in patients suffering from somatoform or conversion disorders [16]. The clinical features described in the case fit more with an episode of a cerebrovascular nature, in particular, given the reduced duration of the symptoms, it seems likely to be a transient ischemic attack.

### 3.1. Transient Ischemic Attack from Patent Oval Foramen

Paradoxical embolism is the most common proposed mechanism at the base of cerebrovascular episodes caused by patency of foramen ovale [17]. This diagnosis therefore seems unlikely, given the exclusion of possible outbreaks at the base of an episode of paradoxical embolism (the echocardiographic examination was normal, as color doppler ultrasound of the supra-aortic vessels, of the splanchnic vessels and of the lower limbs). Moreover, thrombophilic screening excluded hereditary thrombophilic diseases. 

### 3.2. Transient Ischemic Attack of Infectious Etiology

The fact that patient’s symptoms were strongly suggestive of transient ischemic attack (as described in the previous paragraphs) and the exclusion of possible causes of thromboembolism, in association with the positivity of specific IgG and the weak positivity of the pharyngeal swab before entry into our institute (which suggest a recent infection) have led us to the diagnosis of an acute complication attributable to SARS-CoV-2 infection. It should be emphasized that the patient had no risk factors or familiarity for neurological diseases.

Divyani et al. have summarized in a review the possible post-infectious neurological sequelae from SARS-CoV-2, in particular sensory alterations, various degrees of CNS abnormalities up to necrotizing encephalopathy and also stroke are cited. Immune-mediated mechanism is certainly the most cited, although new evidence is emerging regarding the direct neuroinvasivity of coronaviruses. Among the complications, long-term sequelae such as neurodegenerative diseases were also mentioned [3].

In another review, Karimi et al. reiterated how all types of coronavirus can be physiologically related to acute cerebrovascular episodes. In this paper, the proposed mechanism is related to the inflammatory state given by SARS-CoV-2 systemic infection and, in particular, by the consequent state of hypercoagulability [4].

Azorin et al. proposed a multicenter study on neurological complications of SARS-CoV-2 infection. In the paper a total of 233 cases were submitted, including 74 different combinations of manifestations. Stroke was actually the most frequent (27% of neurological sequelae) [18].

Koh et al. reported a study focused on the neurological complications of COVID-19 patients diagnosed in Singapore. Among these, in addition to stroke, some cases of TIA are mentioned. In the cases in question, no pediatric patients experienced neurological complications attributable to the infection [19]. 

In 2020 Mantero et al. reported a case of a 47-year-old patient with a severe form of SARS-CoV-2 and cardiovascular comorbidities who developed an episode of TIA. The patient experienced an episode of left hand paresthesia and loss of vision (amaurosis fugax); these symptoms lasted 5 min, and on the next day he experienced two similar episodes of longer duration (15 and 30 min) [20]. In this case, the sensory symptomatology was similar to our patient’s but it was associated with visual alterations and not to aphasia or other speech alterations. Clearly, this case is different, being an adult patient with a severe form of SARS-CoV-2 infection.

There are few papers in which the topic of neurological complications from SARS-CoV-2 infection in pediatric age are investigated. Singer et al. published a review in which several articles were analyzed [21]. These works reported complications in patients aged 0 to 24 years. Among these, stroke is mentioned; however, like the other sequelae, it is described only in patients with multisystem inflammatory form. 

Ray et al. conducted a national cohort study in the United Kingdom (UK) in 2021, in which pediatric neurologists were invited to report cases of neurological complications of SARS-CoV-2 infection in hospitalized children and adolescents, documented by serological positivity or pharyngeal swab. Among 1334 children hospitalized with COVID-19, 52 cases were collected giving an estimated prevalence of 3.8 cases per 100 patients. The cases were divided according to the presence of primary neurological symptoms or associated with a multi-systemic inflammatory infectious form. In the first group, status epilepticus, encephalitis, Guillain-Barré syndrome, acute demyelinating syndrome, chorea, psychosis, isolated encephalopathy, and only one case of transient ischemic attack were reported [22]. Cerebrovascular complications in SARS-CoV-2 infection in our study were not characterized by a multisystemic inflammatory form, hence suggesting that a different mechanisms, probably of direct neuroinvasiveness, may underlie the central nervous system involvement in this disease.

It is now well documented how the virus can access the central nervous system via the hematogenous route (most likely upon disruption of the blood−brain barrier) or via trans-synaptic pathways, such as along the cranial nerves (VII, IX and X) from nasopharyngeal, respiratory, and/or gastrointestinal access points [23,24]. The virus binds to the angiotensin-converting enzyme receptor 2 (ACE2), present in human adult and fetal brain, with highest expression in the pons and medulla oblongata [25]. Moreover, ACE2 is expressed in pericytes, cells that play key roles in the cerebral microvasculature [26]. SARS-CoV-2 may also bind to the neuronal adhesion molecule neuropilin 1 (NRP1), a glycoprotein essential for normal nervous and cardiovascular system formation and function in vertebrates (also expressed in endothelial cells and neurons). NRP1 plays essential roles in cerebral vasculogenesis, among other things [27]. Furthermore, loss of ACE2, that occurs in cells infected with SARS-CoV-2, increases the risk of vasoconstriction, procoagulation, and inflammation from unopposed angiotensin II [28]. 

Although there are few cases with severe neurological manifestations (encephalitis, meningitis, and intracranial ischemia/hemorrhage) in which a positive polymerase chain reaction (PCR) for the virus has been documented on autopsy brain specimen—and almost none in cerebrospinal fluid—direct replication of the virus in the central nervous system nevertheless appears plausible [23,29,30].

## 4. Conclusions and Take-Home Message

The case we report concerned a patient with asymptomatic infection—apart from a short headache episode two days prior to the appearance of neurological symptoms—thus configuring an absolutely peculiar situation. To our knowledge, no case of transient ischemic attack has been reported in a pediatric patient with SARS-CoV-2 infection, and with this unique onset symptomatology (aphasia and sensory disorders), except for children with underlying risk factors such as Moyamoya syndrome [31]. It is essential that the medical community, starting with family pediatricians, is sensitized on the topic of short- and medium-long term sequelae of SARS-COV2 infections. The case described suggests that, even in the case of asymptomatic infection, it is necessary to pay attention to the appearance of alarm signals, such as neurological symptoms of a dubious nature, i.e., language disorders, alterations in the state of consciousness, focal deficits, and visual disorders. In that case, access to an emergency area is indicated, followed by specialist evaluation by a pediatric neurologist and, possibly, also by a pediatric infectious disease consultant. The table below (Table 2) schematizes the investigations that can be carried out in case of neurological symptoms attributable to SARS-CoV-2 infection (documented with swab and/or positive serological investigation).

It could be important to deepen this topic with further studies, in order to improve our knowledge on the complications of SARS-CoV-2 infection, whether symptomatic or asymptomatic. This aspect is even more important in the pediatric field, given the possible consequences on the neuro-psycho-motor outcome of affected children. However, the evidence on this theme is still too limited to allow us to confirm a real correlation with long-term sequelae.

## Figures and Tables

**Figure 1 children-09-00983-f001:**
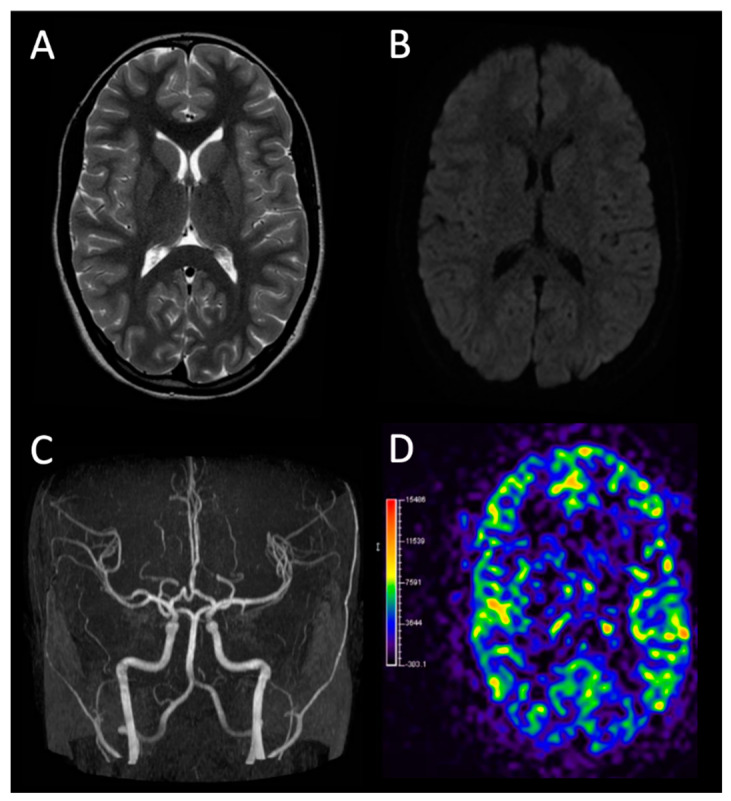
Brain MRI and MR angiography of the patient at clinical onset. (**A**) Axial T2-weighted image at the level of the basal ganglia and (**B**) corresponding axial diffusion weighted imaging (DWI) demonstrate normal parenchymal findings. (**C**) MR angiography reveals normal flow signal at the level of the intracranial arteries. (**D**) Axial arterial spin labeling (ASL) colored map shows normal brain perfusion.

**Table 1 children-09-00983-t001:** Prospectus with blood tests. The values outside the reference range for that parameter are highlighted in red.

*Blood Tests*			
	Units of Measurement	Normal Reference Range	Result
**AUTOANTIBODIES (Ab)**			
Anti Cardiolipin Ab		0–0	5.3
Anti beta2-Glicoprotein 1 Ab		0–0	2
Anti Transglutaminase Ab	UR/mL	0–19	<2
**SERUM EXAMINATION**			
Sodium	mEq/L	135–145	138
Potassium	mEq/L	3.5–5.1	4.3
Calcium	mEq/L	4.05–5.2	4.75
Chlorine	mEq/L	98–110	105
Bicarbonates	mEq/L	20–25	21.9
Anionic gap		7–14	11
Azotemia	mg/dL	15–40	27
Blood glucose	mg/dL	60–100	88
Magnesium	mg/dL	1.7–2.55	2.05
Phosphorus	mg/dL	3–5.6	3.84
Creatinin	mg/dL	0.4–1.2	0.72
AST transaminase	U/L	0–35	16
ALT transaminase	U/L	0–35	8
Gamma-glutamyl transferase (GGT)	U/L	8–32	8
Alkaline phosphatase	UI/L	57–254	110
Conjugated bilirubin	mg/dL	0–0.3	**0.61**
Unconjugated bilirubin	mg/dL	0–99.99	1.79
Total bilirubin	mg/dL	0–1	**2.40**
Triglycerides	mg/dL	30–160	70
Total cholesterol	mg/dL	80–180	128
HDL cholesterol	mg/dL	65–100	69
LDL cholesterol	mg/dL	<100	59
Homocysteine	umol/L	0–10	8.5
Beta2-microglobulin	mg/L	0–2.2	1.9
Hemolysis index		0–0	absent
Jaundice index		0–0	absent
Lipemic index		0–0	absent
**COAGULATION STUDY**			
Prothrombin activity	%	63–129	100.7
Prothrombin activity		0.74–1.25	0.990
Partial thromboplastin ratio		0.86–1.23	1.03
Partial thromboplastin time	s.	23.2–33.2	27.8
Fibrinogen	mg/100mL	180–350	238
Antithrombin	%	75–125	109
D-dimer	mg/L FEU	0–0.55	<0.55
C protein	%	70–140	103
LAC		0–1.3	1.20
Factor VIII activity	%	50–150	124.75
VON WILLEBRAND factor activity (Ristocetinal cofactor)	%	46–173	68
Mutated FII			Absent
Leiden FV			Absent
VON WILLEBRAND factor antigen	%	50–160	62.36
**BLOOD COUNT**			
Hematocrit	%	38–46	40.4
Hemoglobin	g/dL	11.5–16.5	14.2
Erythroblasts	%	0–0	0.0
Erythroblasts	×10^3^/uL	0–0	0.0
Red blood cells	×10^6^/uL	3.9–5.6	4.52
White blood cells	×10^3^/uL	4–9.8	8.06
MCHC	g/dL	32–36	35.2
MCH	pg	27–32	31.4
MCV	fL	82–96	89.3
MPV	fL	7–12	9.7
Platelets	×10^3^/uL	150–450	167
RDW	%	11.5–16	13.0
Basophils	%	0–1	0.3
Basophils	×10^3^/uL	0–0.08	0.02
Eosinophils	×10^3^/uL	0–0.62	0.22
Eosinophils	%	0–9.3	2.7
Lymphocytes	×10^3^/uL	1.37–6.81	1.69
Lymphocytes	%	20.7–50.4	21.0
Large unstained cells (LUC)	×10^3^/uL	0–0.24	0.12
Large unstained cells (LUC)	%	0–3.4	1.5
Monocytes	×10^3^/uL	0.24–0.71	0.39
Monocytes	%	3.9–9	4.9
Neutrophils	×10^3^/uL	2.1–6.43	5.61
Neutrophils	%	35.5–70.8	69.6
**INFLAMMATION INDICES**			
C reactive protein	mg/dL	0–0.46	<0.46
**HORMONAL TESTS**			
FT4	pg/mL	9.3–17	11.4
TSH	uU/mL	0.2–4.2	1.660
**IMMUNITY STUDY**			
Serum IgA	mg/dL	70–400	**38**
Serum IgG	mg/dL	700–1600	1115
Serum IgM	mg/dL	40–230	69.0
Serum IgM anti-2019-nCoV	AU/mL	0–1	<1
Serum IgG anti-2019-nCoV	AU/mL	<1.1	**18.0**

**Table 2 children-09-00983-t002:** Diagnostic tests to be carried out in case of neurological symptoms in patients with SARS-CoV-2. Echocardiogram with bubble study is indicated in patients with suspected cerebrovascular events caused by patent foramen ovale.

Blood Exams	Instrumental Examinations
Complete blood count and leukocyte count	Brain MRI
Complete serum examination with inflammatory indices	Electroencephalogram
Indices of thyroid function and celiac disease screening	Electrocardiogram and echocardiogram
Extended study of coagulation	Ecocolordoppler exam of lower limbs, abdomen vessels and supra-aortic trunks
Anti-phospholipid autoantibodies	Echocardiogram with bubble study

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
