# Peer review of "An Atypical Case of Aphasia: Transitory Ischemic Attack in a 13-Year-Old Patient with Asymptomatic SARS-CoV-2 Infection"

_children, 2022, doi:10.3390/children9070983_

Round 1
Reviewer 1 Report
In the manuscript entitled “An atypical case of aphasia: transitory ischemic attack in a 13-year-old patient with asymptomatic SARS-CoV2 infection” the authors presented a case of an asymptomatic SARS-CoV-2 infected 13-year-old female patient with an acute episode of aphasia accompanied by left-arm dysesthesia. This is an interesting and well-written manuscript. However, the evidence provided by the authors doesn’t confirm any strong correlation between asymptomatic SARS-CoV-2 infection and the transitory ischemic attack observed in this patient. This is one of the major concerns that need to be addressed before the manuscript can be considered for publication. Authors need to first justify the strong correlation between asymptomatic SARS-CoV-2 infection and the neurological symptoms observed in this 13-year-old patient and also need to mention how it can be considered a special case by pointing out the uniqueness and novelty of the case presented in this study. It would be great if the authors can describe the possible reason behind the asymptomatic SARS-CoV-2 infection leading to neurological aphasia-like symptoms. The authors also need to explain a little bit about the aphasia and TIA followed by cases observed in the patients suffering from SARS-CoV-2 infection. Although some reports have been discussed under the differential diagnosis and discussion section (subheading 3.2), it needs to be written in the form of a connecting story and how it is correlated with the current study.
Author Response
Dear rev.1
Thank you for your valuable comments and remarks, which have certainly enabled us to improve the quality of our paper.
Regarding the issues you raised:
We have reported in the differential diagnosis section the reasons why we believe, in agreement with the opinion of our specialists, that the symptoms presented by the patient are strongly attributable to the effects of SARS-COV2 infection. The symptomatology described is strongly suggestive of transient ischemic attack; in addition, the girl presented an excellent state of health (with no risk factors), and all other possible causes underlying the symptoms were ruled out by the examinations we performed. In our opinion, this case is unique since - especially among adult patients - episodes of cerebrovascular events are mainly reported in forms of disease with multisystem inflammation. There are really few cases in the literature of patients with symptoms similar the ones presented by our patient in non-severe infection; to our knowledge no case of transient ischemic attack has been reported in a paediatric patient, and none with this symptomatology (aphasia and sensory disorders). We have reported in the revised version of the paper, as requested, what we think are the underlying causes of virus-induced neuronal damage even in the absence of inflammatory involvement (in agreement with other studies). As requested, bibliographic references were definitely expanded in the introduction of the paper so as to construct a story connected with the presented case.

Reviewer 2 Report
The authors report an atypical and temporary case of aphasia, attributable to the effects of Covid-19 positivity. I suggest the following comments:
1)Introduction: it would be necessary to expand the introduction with studies on the effects of Covid-19 from a neurological and neuropsychological perspective in the paediatric setting.
2)Table 1: It would be interesting to table the data of the neuropsychiatric tests that were carried out to verify the presence or absence of aphasia
Author Response
Dear rev.2
Thank you for your valuable comments, which allowed us to improve the quality of our paper and our understanding of COVID-related neurologic complications.
Regarding the questions you raised:
1) As requested, the introduction has been expanded with several references in the literature pertaining to the neurological consequences and neuropsychiatric outcome of SARS COV2 infection in the paediatric setting
2) Upon the patient's arrival at our emergency department, at the time of first neurology consult, the symptoms had completely disappeared, and no specific tests with neurological scores for aphasia were performed. We added this piece of information in the revised version of the manuscript.
Moreover, as you have requested, we have made linguistic edits to our manuscript under the supervision of a native English-speaking colleague.

Reviewer 3 Report
Current study introduces an interesting case of a 13-year-old patient with asymptomatic SARS-CoV2 infection who suffered from a transitory ischemic attack.
I appreciate the novelty of the case, along with the fact that such evidence with regard to COVID-19 comorbidity is relatively scarce in existing literature, especially for pediatric populations. Differential diagnosis is well justified.
Perhaps, some information with regard to language and/or neuropsychological assessment, would be beneficial to understand if any cognitive deficits were apparent after the transitory ischemic attack.
Moreover, I would appreciate if the authors could include some images from structural mri.
Author Response
Dear rev.3
We are glad you appreciated the novelty of our case and we thank you for your valuable remarks which have allowed us to further improve the quality of our work.
We have added information about the patient's language and neuropsychological assessment to the text to improve the understanding of the patient's cognitive aspect after the event.
Moreover, as you requested, we added images regarding the patient's brain MRI.

Round 2
Reviewer 1 Report
In the manuscript entitled “An atypical case of aphasia: transitory ischemic attack in a 13-year-old patient with asymptomatic SARS-CoV2 infection” the authors presented a case of an asymptomatic SARS-CoV-2 infected 13-year-old female patient with an acute episode of aphasia accompanied by left-arm dysesthesia. I appreciate the author’s detailed answers and changes within the previously submitted version of the manuscript. Now, the authors were able to clearly address all the comments raised in this study and the suggested modifications have been taken into account in improving the quality of the article. The revised version of the manuscript is clear, concise, and well written.
Minor Comments:
Authors need to follow the same format for naming the SARS-CoV-2 virus within the manuscript.
Reviewer 2 Report
I consider the manuscript with the changes made to be valid for publication